# SemNFT: A Semantically Enhanced Decentralized Middleware for Digital Asset Immortality

## ABSTRACT

Non-Fungible Tokens (NFTs) have emerged as a pivotal digital asset, offering authenticated ownership of unique digital content. Despite it has gained remarkable traction, yet face pressing storage and verification challenges stemming from blockchain's permanent data costs. Existing off-chain or centralized storage solutions, while being alternatives, also introduce notable security vulnerabilities. We present SemNFT, an innovative decentralized framework integrated with blockchain oracle middleware services, addressing these persistent NFT dilemmas. Our approach compresses NFT source data into compact embeddings encapsulating semantic essence. These arrays are stored on-chain, while facilitating reliable decentralized image reconstruction and ownership verification. We implemented ERC721-compliant smart contracts with supplementary functionalities, demonstrating SemNFT's seamless integrative capabilities within the ecosystem. Extensive evaluations evidence marked storage optimizations and preservation of requisite visual fidelity by comparison with existing solutions. The proposed SemNFT framework marks a significant advancement in holistically confronting rising NFT storage and verification challenges without compromising decentralization. It substantively propels the meaningful evolution of NFT infrastructure to achieve digital asset immortality.

## CCS CONCEPTS

• **Computing methodologies** → **Image compression**; **Neural networks**; • **Security and privacy** → **Digital rights management**; • **Applied computing** → *Media arts*.

## KEYWORDS

Blockchain, Decentralized System, Middleware, Autoencoder, Non-Fungible Token, Deep Learning, Smart Contract

**ACM Reference Format:**
Anonymous Author(s). 2024. SemNFT: A Semantically Enhanced Decentralized Middleware for Digital Asset Immortality. In *Proceedings of the 32nd ACM International Conference on Multimedia (MM '24), October 28-November 1, 2024, Melbourne, VIC, AustraliaProceedings of the 32nd ACM International Conference on Multimedia (MM'24), October 28-November 1, 2024, Melbourne, Australia.* ACM, New York, NY, USA, 9 pages. https://doi.org/10.1145/3664647.3681114

## 1 INTRODUCTION

Non-Fungible Tokens (NFTs) [1, 21, 39] are a type of virtual token that utilizes blockchain [12] technology to authenticate decentralized digital asset ownership. NFTs can represent any unique digital asset, such as images, music, videos, in-game items, domain names, collectibles, and more. One of the key advantages of NFTs is their ability to ensure the authenticity, ownership, and transferability of digital assets, while also reflecting the asset's scarcity and cultural value [15, 44]. The market for NFTs has experienced explosive growth in recent years; according to data from Forbes, the total transaction volume for NFTs exceeded $23 billion in 2021 [4]. Furthermore, the NFT market has become the most gas-consuming Ethereum contract. For instance, popular NFT trading platforms like Opensea[1] rank among the top in Etherscan[2], consuming 20% of the entire Ethereum network's gas fees.

The surge in the number of NFTs has raised concerns about their secure and reliable storage [19, 46]. The high costs associated with data storage on public blockchains like Ethereum [12] are well recognized. The inherent design makes permanent data storage on Ethereum particularly cost-inefficient due to substantial gas fees, which also contribute to the blockchain's load. As per existing metrics, to store 1 KB of data on Ethereum, around 640,000 gas is needed. Taking an earlier gas price of 12 Gwei [22] into account, this translates to approximately $24 USD, going by recent exchange rates. This cost escalates drastically when it comes to storing larger files such as images or webpages, with expenses potentially rising to thousands of dollars. Therefore, how to reliably store these files on a high-cost public chain is a topic worthy of research.

According to data from [26], approximately 9% of NFTs are stored on blockchain, another 55% are stored on private servers, and the remaining 36% are stored on the InterPlanetary File System (IPFS) [6]. For NFTs stored on-chain, these are generally generative art pieces, such as those from Art Blocks[3], where the generating script is stored on the blockchain. However, more than 90% of NFTs require the storage of their own metadata. The 40% of NFT-related artworks stored on off-chain or centralized private servers pose significant security risks and reliability concerns. If the server shuts down, the NFT will no longer point to the artwork or file but to a broken link that cannot be accessed. Therefore, decentralized storage protocols like IPFS and Arweave [48] are the current solutions for most NFT projects. These solutions offer more affordable, reliable, and flexible decentralized storage services and can also interoperate with Ethereum. However, this storage method still has a high likelihood of causing users to lose their NFT ownership. For example, if no one pays to pin an image on IPFS, the system will eventually delete it during routine cleanups to reduce redundant data. However, most artists or collectors do not use paid services to

---

[1]https://opensea.io/
[2]https://etherscan.io/
[3]https://www.artblocks.io/

pin artworks, considering the technical barriers and financial costs. Between June and December 2021, "3.91% of assets and 9.04% of metadata records hosted on IPFS" disappeared [19]. Failed metadata and assets subsequently cannot be matched and verified with the user's wallet address. The loss of ownership also renders the digital assets valueless, resulting in significant losses for consumers.

**Motivation**. The crux of the challenge in the NFT landscape lies in addressing current storage inefficiencies and trust dilemmas. There's a pressing need to employ a more advanced compression technique that enables storing image data on the blockchain with minimized space and gas costs. Concurrently, a clear shift is discernible within the community: it's essential to shift towards a more decentralized verification process, diminishing the heavy reliance on and trust issues with private servers or IPFS. Furthermore, it's imperative that any innovative framework developed is compatible and can easily merge with the prevailing NFT protocols or standards, ensuring the holistic advancement of the NFT ecosystem.

**Approach**. In light of these challenges, drawing from the realms of deep learning, blockchain, and cryptography, we propose a decentralized NFT storage and verification framework grounded in autoencoder and oracle middleware, denoted as SemNFT. While our primary focus in this paper is on image-based NFTs, we postulate that the core tenets of our framework can be extended to any types of NFTs storage and verification context. Initially, our approach capitalizes on self-supervised learning to distill the semantic essence of NFT images, striving for embeddings that encapsulate core features while balancing storage costs and reconstruction fidelity. With the image semantics inscribed on the blockchain, the onus of NFT ownership verification shifts to the compact autoencoder model, circumventing the bulky image data. This model, distributable and storable across diverse locations, aligns its verification with the image reconstruction on the blockchain oracle or the hash value of the smart contract, sidestepping the problems of IPFS or private server dependencies. Conclusively, our SemNFT framework is architecturally congruent with established NFT protocols, such as ERC721 [21], and introduces supplementary functionalities, underpinning its seamless integration into the prevailing ecosystem.

**Contributions**. Our contributions can be concluded as follows:

- **Framework**. We introduce a novel NFT storage framework that foregrounds efficiency, decentralization, and compatibility, striving to achieve digital asset immortality. This framework establishes a robust connection between user wallets and NFT assets, effectively addressing concerns from both users and the broader community regarding NFT ownership.
- **Implementation**. We have created a demonstration for this NFT framework, showcasing its seamless integration capabilities with existing NFT protocols, thereby highlighting the protocol's ease of use and scalability.
- **Evaluation**. We conducted performance evaluations of the proposed framework in terms of reconstruction results and storage costs under different parameters. Compared to existing solutions, we demonstrated its feasibility and efficiency.

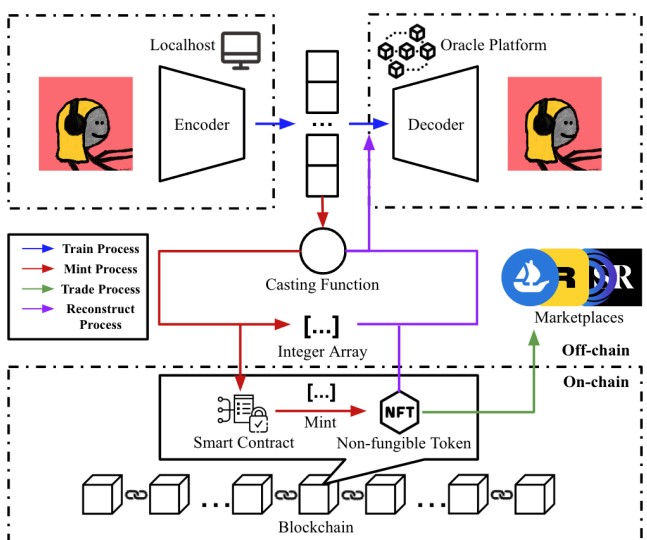

**Figure 1: The overview architecture of SemNFT**

## 2 PRELIMINARY

### 2.1 NFT Infrastructure

**NFT Protocols and Interfaces.** In the Ethereum ecosystem, there are currently several predominant contract protocols for implementing NFTs through interfaces. Notably, ERC721 [21] enables the minting of a single NFT, whereas ERC1155 [39] can represent varying values, depending on whether they are fungible, semi-fungible, or non-fungible [1]. Other NFT protocols like ERC998 [37], ERC2981 [11], and ERC3525 [45] improve NFTs' compatibility and interoperability [28, 46]. As of today, digital assets embodying transferrable rights, minted via these protocol interfaces, have proliferated across various domains, including art, in-game items, investment markets, collectibles, and music [2].

**NFT Asset Storage.** In the domain of NFT resource storage, two primary storage methods prevail: centralized storage and decentralized storage. Centralized storage is typically employed within NFT marketplaces, such as Opensea, Nifty Gateway[4], and Rarible[5] [46]. By connecting to resources stored on centralized servers, this approach facilitates faster network transmission of resources. However, it also implies that NFT owners relinquish some degree of control over their NFTs. To ensure decentralized storage, numerous decentralized storage systems have emerged, including IPFS [6], Swarm [42], , and Arweave [48], as elucidated in [8, 18]. Given that physical devices entail operational costs, file storage cannot indefinitely provide services to users free of charge. So, the Incentive Layer has surfaced above the Storage Layer, exemplified by protocols like Filecoin [5].

### 2.2 Blockchain Oracles

Blockchain oracles shown in Figure 3 are introduced as middleware services. The blockchain environment is considered isolated and independent relative to the external world [43]. Consequently, when smart contracts require Oracles to access real-world data

---

[4]https://www.niftygateway.com/
[5]https://rarible.com/

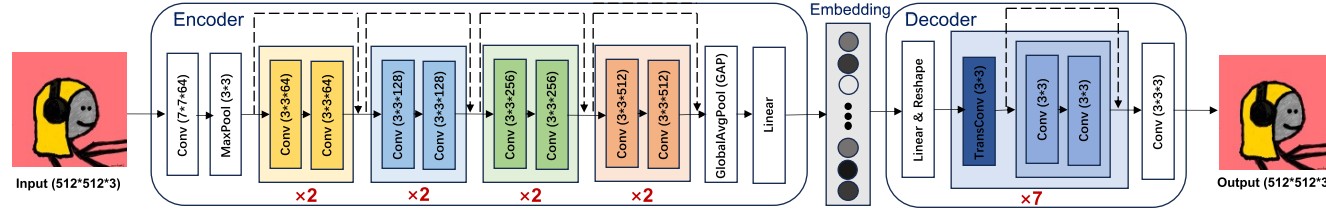

Figure 2: The proposed architecture of the autoencoder

[51], particularly with the emergence of Decentralized Applications and platforms [50]. Oracle services can be categorized into several types, including Software Oracles, Hardware Oracles, Human Oracles, Computation Oracles, Inbound/Outbound Oracles, Contract-specific Oracles, and Consensus-based Oracles [7, 13]. The currently prominent blockchain oracle platforms include Chainlink [10] and API3 [9]. In this study, to address the limitations of blockchain computational capabilities and gas limits, we adopt the pattern of Computation Oracles to perform off-chain computations and derive the expected results.

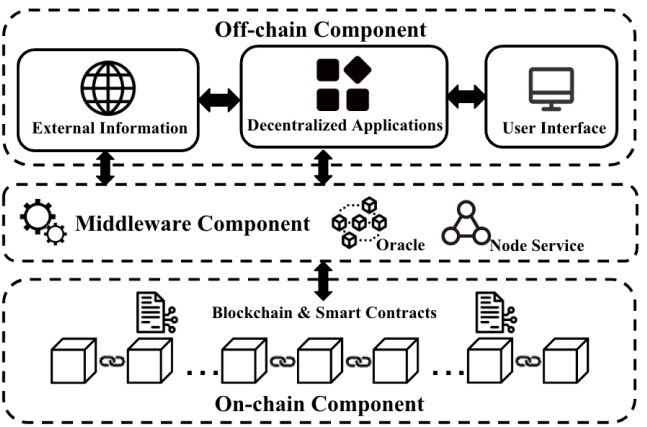

Figure 3: The oracle is a middleware component between off-chain and on-chain environments.

## 3 RELATED WORK

**Autoencoder Compression**. Several studies [16, 17, 20] have explored and implemented data compression techniques for images using various autoencoder architectures. [23, 53] has combined recurrent structures with autoencoders to achieve data compression for videos. In addition, researchers have employed deep autoencoders and variational autoencoders to compress 3D models represented as point clouds and voxels [36, 52].

**Compression related to blockchain**. Over the last few years, the emergence of storing and compressing data on the blockchain has become prominent. In addition to directly compressing blockchain blocks and optimizing the chain's structure [54], there are other methods and applications for data compression:

SCC [30] is storage compression consensus algorithm which compresses a blockchain in each device to ensure the storage capacity. SELCOM [29] is a selective compression scheme using a checkpoint-chain to prevent the accumulation of compression results. These

solutions address the issue of insufficient storage capacity in lightweight Internet of Things (IoT) devices and have demonstrated promising performance in their respective experiments. However, they are only suitable for consortium chains utilizing consensus algorithms such as Practical Byzantine Fault Tolerance (PBFT), and are not applicable to public chain systems such as Ethereum or Polygon. DVSSA [40] introduces a sequential aggregate signature scheme with a designated verifier. By sequentially aggregating the signatures of all participants, this approach reduces the size of the signature stored on the blockchain to that of a single person's signature, resulting in significant storage space savings. However, it is important to note that this algorithm compresses only the signature information and does not store data on the blockchain itself. Accessing the data still requires centralized cloud-based storage.

**Computation Oracles**. There are many implementations and applications concerning Computation Oracles. In the realm of outsourced polynomial computation, [24] combined blockchain oracles to propose a novel computational scheme. In the field of trust management, [31] contributed trustless smart oracles to the Fog Computing Platform. In the field of semantic communication, [35] alleviates communication and storage pressures in blockchain data exchange to prevent network latency from information overload. What's more, the consensus problem would happen when outputs from nodes are slightly different. The developer can customize the consensus mechanism by aggregating outputs and confirming the final result. As for outliers, reputation-based oracles reduce node reputation, and stake-based oracles impose economic penalties.

## 4 METHODOLOGY

### 4.1 Architecuture

The architecture of the protocol can be divided into two parts, the **off-chain part** and the **on-chain part**, as shown in Figure 1. In the off-chain, the primary focus involves two key tasks: training the autoencoder model and downcasting the float array. The autoencoder model training process plays a vital role in data compression and feature extraction, while the downcasting of the float array aims to convert floating-point numbers to integers for subsequent operations. On the other hand, the on-chain part is primarily dedicated to the minting of NFTs from the integer array, which are subsequently stored and managed on the blockchain. This process enables the unique identification and ownership tracking of individual NFTs within the decentralized ledger system.

In this study, the diagram depicts various processes. The blue arrows symbolize the **Train Process**, which involves the training of an autoencoder model for data compression and feature extraction. The red arrows represent the **Mint Process**, where NFTs are

generated and created from an integer array, subsequently stored on the blockchain. The green arrows indicate the **Trade Process**, illustrating user-driven activities such as trading, exchanging, and transferring NFTS on the blockchain. The purple arrows signify the **Reconstruct Process**, which involves users actively reconstructing NFT images from the data stored on the blockchain.

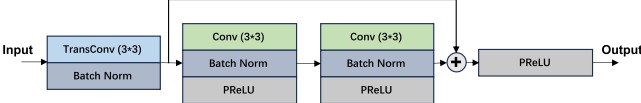

**Figure 4: Detailed structure of the designed upsampling block**

## 4.2 Neural Network Design

Based on the designed architecture, we can broadly utilize any neural network structure specifically tailored for image autoencoders in SemNFT framework. The purpose of using the autoencoder is to harness self-supervised learning to extract semantic features from the NFT image source data, thereby enhancing storage efficiency and ensuring restorability for ownership verification. In the proposed framework, we adopted a convolutional autoencoder architecture, synthesizes the power of established convolutional neural networks (ResNet-18)[25] and innovative deep residual upsample blocks. The architecture presents detaily in Figure 2, formulated as a function mapping $X \rightarrow \hat{X}$, where $X$ and $\hat{X}$ are the original and reconstructed images respectively, is delineated into three distinct modules: Encoder $E$, Embedding, and Decoder $D$.

Given an input image $X$, the encoder module $E$ harnesses the ResNet-18 architecture, renowned for its resilience to vanishing gradient issues and adept feature extraction capabilities. Mathematically, the encoder is a function

$$E(X) = F(X; \Theta_E) \tag{1}$$

where $F$ represents the mapping function defined by the sequential application of convolutional, normalization, and activation layers, and $\Theta_E$ represents the parameters of these layers. The outcome, $Z_E$, is a condensed feature representation, primed for further compression and feature abstraction in the embedding module.

In order to further investigate the impact of Embedding Size on image reconstruction quality and on-chain gas consumption, and to offer the NFT issuing party the flexibility to customize the Embedding Size as a trade-off between gas cost and image reconstruction benefits, we have introduced an Embedding Block with configurable Embedding Size. This block is inserted between the Encoder and Decoder and utilizes a fully connected layer (Linear) to accommodate the Embedding Tensor. Considering that the Input Size and Output Size of this block are both 128, it is recommended to set the Embedding Size to a value not exceeding 128.

The decoder module $D$, with its deep upsampling residual blocks, reconstructs a high resolution image $\hat{X}$ from the embedding $Z$. Formally, the decoder is a function

$$D(Z) = I(Z; \Theta_D) \tag{2}$$

where $I$ represents the mapping function facilitated by the sequential application of deep upsampling residual blocks, and $\Theta_D$ denotes

the parameters of the decoder. The detailed visualization of upsampling residual block $F$ is shwon in Figure 4, which mathematical formulation can be expressed as:

$$F(x; \Theta_F) = \mathcal{P}(F_{\text{conv}}(x; \Theta_{\text{conv}}) + F_{\text{up}}(x; \Theta_{\text{up}})) \tag{3}$$

where the parameters $\Theta_F$ in the residual block comprise both $\Theta_{\text{conv}}$ and $\Theta_{\text{up}}$. Consequently, the full decoder $D$, which is a sequence of these residual blocks, will possess parameters $\Theta_D$ that embody the collection of all $\Theta_F$ for each block in the series. $\mathcal{P}(x)$ is the PReLU activation function, compared to ReLU, PReLU prevents neuron death in the network when encountering negative values and also improves image retrieval [16, 38]. The mathematical formula is:

$$\mathcal{P}(x) = \max(0, x) + \alpha \min(0, x) \tag{4}$$

with $\alpha$ as a hyperparameter, and we use $\alpha = 0.25$.

To ensure the fidelity of the reconstructed image $\hat{X}$ to the original $X$, a Mean Squared Error (MSE) loss function is employed to guide the optimization of the model parameters during training. Formally, the loss function $L$ is given by:

$$L(X, \hat{X}) = \frac{1}{N} \sum_{i=1}^{N} \left(X_i - \hat{X}_i\right)^2 \tag{5}$$

where $N$ is the total number of pixels in the image, and $X_i$ and $\hat{X}_i$ denote the original and reconstructed pixel values, respectively. The minimization of $L$ ensures the model learns to preserve critical information through the encoding and decoding processes.

## 4.3 Casting

In Solidity, while the data type Fixed Point Numbers can be declared to represent floating-point numbers, it is important to note that Fixed Point Numbers are not fully supported in Solidity[6]. Although they can be declared, they are currently not fully functional for assignments and operations within the language. As a result, alternative approaches must be explored to address the limitations in representing and working with floating-point numbers in Solidity.

Currently, this limitation in Solidity is not suitable for the storage and subsequent operations of floating-point arrays in this study. However, drawing inspiration from the concept of fixed-point numbers, we truncate the integer and fractional parts of floating-point numbers, referring to this process as the "Casting Function." We utilize the Int8 data type to represent the integer part of floating-point numbers and employ one of four different data types, namely Truncate, UInt8, UInt16, or UInt32, to store the fractional part.

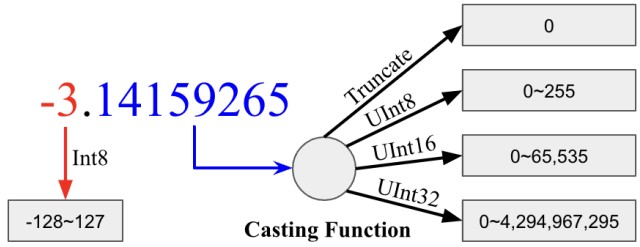

**Figure 5: The overview of casting methods.**

---

[6]https://docs.soliditylang.org/en/latest/types.html#fixed-point-numbers on 2023-09-23

In the subsequent experiments, we utilize the aforementioned four casting functions to process the fractional part of floating-point numbers. The processing procedure is illustrated in Figure 5. Subsequently, we will conduct a comparative analysis of image quality among the reconstructed images from raw floating-point numbers, Truncate, UInt8, UInt16, and UInt32 embeddings.

Due to the static typing nature of Solidity, it is unable to store elements of mixed types. Consequently, we store the integer and fractional parts of embeddings in two separate mapping arrays. Subsequently, we reconstruct the original floating-point numbers by employing token ID-based lookup procedures.

This design benefits from the characteristics of Solidity packing[7], where we arrange variables of the same type consecutively in three arrays. This allows Solidity to efficiently pack the variables, leading to significant savings in on-chain data storage and gas consumption.

## 4.4 Contract Design

In this protocol, smart contracts serve as the definitive entities for determining NFT ownership, initiating NFT transfers, storing NFT image embeddings, and validating the model. We will utilize the ERC721 module[8] from the *OpenZeppelin library*[9] for secure smart contract development to assist in the contract design of SemNFT. By inheriting the features of OpenZeppelin's ERC721 module, the contract design and development for SemNFT become streamlined. This approach enables NFT issuers to focus solely on the embedding-related logic of SemNFT, alleviating the need to be concerned with designing fundamental NFT functionalities.

---

**Contract SemNFT is ERC721**

ERC721 **values**;
uint8 public **emb_size**;
string public **model_hash**;
mapping(uint256 => int8[]) public **int_map**;
mapping(uint256 => uint16[]) public **decimal_map**;
address public **oracle_address**;
string public **oracle_job_id**;

ERC721 **functions**;
function **setEmb**;
function **getEmb**;
function **checkModelHash**;
function **requestOracle**;

---

**Figure 6: The brief view of the smart contract with UInt16 decimal casting.**

The design of data structures and functions for smart contracts is depicted in Figure 6. This contract inherits from the ERC721 module, thus inheriting the state and methods from ERC721.

---

[7]In contracts, state variables are efficiently stored in storage, often in a compact manner, where multiple values might share the same storage slot. https://docs.soliditylang.org/en/v0.8.16/internals/layout_in_storage.html
[8]https://docs.openzeppelin.com/contracts/4.x/erc721
[9]https://www.openzeppelin.com

## 4.5 Compression Ratio

In this work, as the dataset of images is divided into two components, namely model parameters and on-chain storage, when model parameters, embedding size, and casting precision are fixed, the total space required for the project remains constant and does not vary with the size of the image dataset. In PyTorch models, parameters are typically represented using the float32 data type, with each float32 occupying 4 bytes.

Therefore, using $D$ to represent the size of the original dataset, $D'$ as the size of the compressed dataset, $N$ as the number of images, $p$ as the number of model parameters, $\theta$ as the embedding size, and $C$ as the size occupied by a numerical value representing the fractional part's casting precision, we can derive the following formulae.

$$D' = 4 \times p + (\text{int8} + C)\theta N \quad (6)$$

The value of parameter $C$ is the size of one integer number occupied in EVM determined by the integer type such as UInt8, UInt16 and UInt32. Therefore, when the other parameters in the formula are determined, the compression ratio ($\eta$) is determined by the size of the original dataset as the following equation:

$$\eta = \frac{D}{D'} \quad (7)$$

For instance, if a 1GB NFT image dataset comprises 10,000 images, the trained model occupies 100MB, and an embedding size of 50 with UInt16 is selected, then after computation, the total file size amounts to 101.5MB, resulting in a compression ratio of 9.85:1. Due to the additional data compression applied to arrays by the EVM, the actual compression ratio is expected to further increase.

## 4.6 NFT Verification

Figure 7 illustrates the NFT ownership verification process of SemNFT in comparison to existing NFTs. Under the current NFT storage scheme, if we need to verify an NFT image with unknown ownership, we must seek the NFT owned by the user in the contract through their wallet address. Subsequently, we retrieve the NFT resource's metadata through the link recorded in the contract and then request the image link in the metadata to obtain and verify the NFT image. If a disconnection occurs during the link request process, the verification process will be interrupted. Even if users store their NFT image source data off-chain, it cannot be matched and verified with assets in the smart contract.

In contrast, within the SemNFT framework, the intrinsic features containing user NFT image semantic information are stored on the blockchain and can be reconstructed into NFT images through the decoder in the autoencoder model, completing ownership verification. The autoencoder model can be regarded as a specialized, customized image decoding protocol, and its accuracy can be verified by the model hash value stored in the smart contract. Therefore, this autoencoder model and reconstructure process can be in the trusted blockchain oracle as the middleware service. Consequently, issues of consumer and community trust and dependece on IPFS or private servers are resolved.

**Figure 7: Ownership verification in existing NFT & SemNFT**

## 5 IMPLEMENTATION

This section will simulate the implementation of data preparation, model training, contract deployment, and embedding on-chain for SemNFT, based on the aforementioned protocol design.

### 5.1 Target Datasets

To validate the universality and applicability of this protocol, we conducted verification using three distinct styles of NFT datasets available on the HuggingNFT dataset collection[10] on HuggingFace: *Cryptopunks*[11], *Boredapeyachtclub (BAYC)*[12], and *Azuki*[13]. Elaborated details of these datasets are presented in Table 1. It can be observed that the envisioned difficulty in feature extraction and reconstruction increases sequentially across these three datasets.

**Table 1: Target Datasets**

| Dataset | Cryptopunks | BAYC | Azuki |
|---------|-------------|------|-------|
| Amount | 10,000 | 9,999 | 10,000 |
| Image Size[14] | 312x312 | 631x631 | 2000x2000 |
| File Size | 12.1 MB | 1.2 GB | 1.11 GB |
| Details | Low | Relatively High | High |

### 5.2 Model Training

**Training Equipment**. The primary device used for training the neural network model is a host with a single GeForce RTX 3090 graphics card, an Intel i9-10900K CPU, a 64 GB memory, and running the Ubuntu 18.04 LTS Bionic Beaver operating system.

**Training Settings**. The training process employs the Adam optimizer with a learning rate set to 0.0001. Additionally, EarlyStopping is implemented with a patience of 10 epochs, allowing for early termination if the validation performance does not improve.

---

[10]https://huggingface.co/huggingnft
[11]https://cryptopunks.app/
[12]https://boredapeyachtclub.com/
[13]https://www.azuki.com/
[14]The image size is the same as that of the image origins in the HuggingFace dataset, and does not represent the actual dimensions of the project images. The actual pixel dimensions for *Cryptopunks* are 24x24, and they are in SVG format with the basic Solidity data structure without ERC721 or ERC1155.

### 5.3 Test Chain Deployment

In the realm of Ethereum, there exist numerous test networks, including Ropsten[15], Kovan[16], Goerli[17], and Sepolia[18], each distinguished by varying Gas Prices and blockchain consensus algorithms. Furthermore, it is noteworthy that some of these test networks may be rendered obsolete over time. In contrast, Polygon's[19] array of test networks currently consists of only one, namely Mumbai[20], characterized by its singularity and commendable stability. Therefore, we have opted to deploy a portion of our smart contracts on Polygon's Mumbai test network. Given that these contracts are authored in Solidity, they are universally compatible with any blockchain supporting the EVM [3].

In the forthcoming experiments, we intend to deploy four different contracts, Truncate, UInt8, UInt16, UInt32, each varying in decimal precision. Furthermore, We will employ the MetaMask browser plugin to interact with these contracts [32], evaluating functionalities, as well as assessing the cost associated with NFT SafeMint and reading embedded data.

## 6 EVALUATION

### 6.1 Embedding Size Factor

We utilized two algorithmic metrics, the Structural Similarity Index (SSIM) and the Peak Signal-to-Noise Ratio (PSNR) [27], to compare the quality of the reconstructed images without using the casting function across different datasets and embedding sizes. In Figure 8, the horizontal axis represents the size of the embedding, while the vertical axis displays the values of the algorithmic metrics.

From the graph, it becomes evident that the *Cryptopunks* dataset, which contains the least amount of detail, consistently exhibits the highest SSIM and PSNR values. Conversely, the *Azuki* dataset, with the most intricate details, generally displays lower SSIM and PSNR values compared to the other two datasets.

Within the range of embedding sizes from 10 to 30, both SSIM and PSNR exhibit slight improvements as the embedding size increases, with the most notable enhancement observed in the *Azuki* dataset. However, beyond an embedding size of 30, SSIM and PSNR show fluctuations with increasing embedding size. This suggests that, at this point, the spatial capacity of the embedding size is sufficient for evaluating the reconstructed image quality.

### 6.2 Casting Precision

Initially, we conducted an assessment of images reconstructed at various levels of precision. Using images reconstructed through the Float method as the benchmark, we proceeded to perform statistical t-tests to compare the SSIM and PSNR of images by all embedding size (range from 10 to 100), and with differing precision levels, employing the Truncate, UInt8, UInt16, and UInt32 methods. The predetermined significance level was set at $\alpha = 0.05$, and the p-values are presented in Table 2 as the reconstructed stability.

---

[15]https://ropsten.etherscan.io/, now deprecated
[16]https://kovan.ethplorer.io/
[17]https://goerli.etherscan.io/
[18]https://sepolia.etherscan.io/
[19]Polygon launched as Matic Network in 2017, and rebranded to Polygon in 2021.
[20]https://mumbai.polygonscan.com/

**Table 2: The comparison of P-values from t-tests, conducted for the SSIM and PSNR between reconstructed images obtained through the Truncate, UInt8, UInt16, Uint32, with Float methods, in significance level of $\alpha = 0.05$.**

| Method | Truncate | | UInt8 | | UInt16 | | UInt32 | |
|---|---|---|---|---|---|---|---|---|
| | SSIM | PSNR | SSIM | PSNR | SSIM | PSNR | SSIM | PSNR |
| **Cryptopunks** | 0.0 | 0.0 | 0.891069 | 0.923751 | 0.999547 | 0.999815 | 0.999933 | 0.999916 |
| **BAYC** | 0.0 | 0.0 | 0.975591 | 0.940597 | 0.999973 | 0.999941 | 0.999998 | 0.999971 |
| **Azuki** | 0.0 | 0.0 | 0.925515 | 0.035697 | 0.999567 | 0.994141 | 0.999998 | 0.999932 |

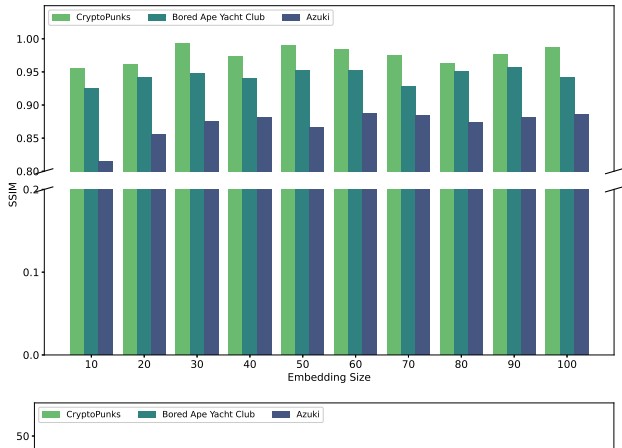

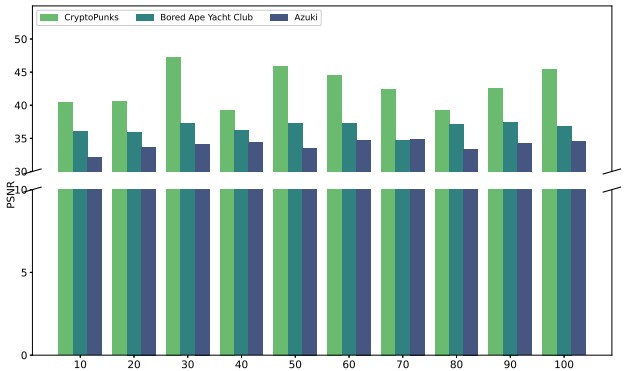

**Figure 8: SSIM and PSNR mean values with the embedding size and different NFT datasets.**

From the experimental results, we observe that for all three NFT image datasets, the p-values for SSIM and PSNR when using the Truncate method are both 0. This suggests that the Truncate method results in significant distortion and divergence from the original image, likely because the fractional parts of the elements in the embedding array, preserved by the current autoencoder model, contain crucial image information. Loss of these fractional parts leads to substantial image distortion. In contrast, for methods other than Truncate, the p-values for the restored images increase as the casting function enhances the precision of float values. The UInt32 method, which achieves the highest precision, yields p-values very close to 1, indicating minimal distortion and better restored stability.

The observed significant differences in the PSNR p-values for the *Azuki* dataset when using the UInt8 method may indeed be influenced by the dataset's inherent characteristics. The dataset's large number of features, uneven distribution of feature types, and rich feature details likely contribute to this discrepancy. For instance, in some *Azuki* images, certain characters may have accessories with intricate pattern designs. When attempting to reconstruct these intricate details, the presence of noise peaks in the PSNR values

might be relatively higher due to the complexity of restoring such fine-grained details accurately.

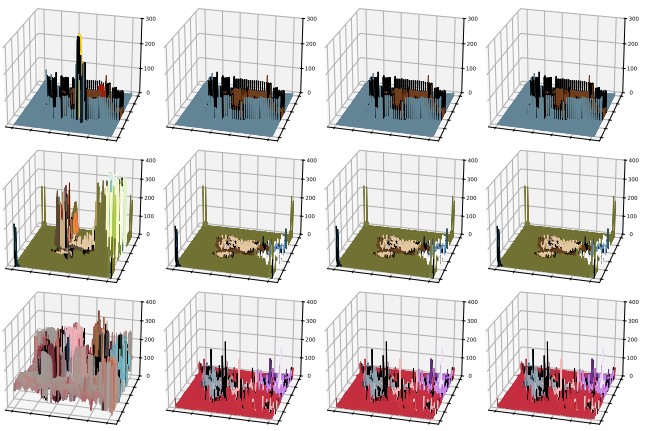

   (a) Truncate     (b) UInt8     (c) UInt16     (d) UInt32

**Figure 9: The comparison among reconstructed by Truncate, UInt8, UInt16, UInt32 in the embedding size of 30 with original images. The datasets, from top to bottom, represent Cryptopunks(#427), BAYC(#79), and Azuki(#1227).**

In the t-test, when the p-value is less than the significance level, it indicates that the difference between the datasets is statistically significant. For the UInt16 and UInt32 precision methods, both the SSIM and PSNR comparisons with Float yield p-values very close to 1. It implies that these precision methods are highly similar to the Float method. Therefore, in practical usage, it is recommended to employ the UInt16 casting methods to ensure the quality of the reconstructed images with lower storage cost, as they provide results that are statistically comparable to those obtained with Float precision. Besides, we use the distance equation to compute the pixel RGB difference between the constructed and the original:

$$d = \sqrt{(p_{red} - p'_{red})^2 + (p_{green} - p'_{green})^2 + (p_{blue} - p'_{blue})^2} \quad (8)$$

The visualization of the individual distance shows in Figure 9.

## 6.3 Transaction Fee

In the context of minting NFTs, in addition to any additional fees that may be set by service providers, the minter is required to cover the Transaction Fee to accommodate the Gas required by the EVM. The concept of Gas operations is formally defined in Dr. Wood's Ethereum yellow paper [49]. Gas costs serve as a mechanism to mitigate Denial-of-Service attacks in the execution of code on the Ethereum platform [41]. The computation of both the Transaction Fee and Gas is outlined as follows, with the Gas Price being subject to dynamic fluctuations based on the congestion level within the

blockchain network [47]. The Gas Used varies according to the complexity of the invoked function and the size of stored data:

$$\text{Transaction Fee} = \text{Gas Used} \times \text{Gas Price} \tag{9}$$

Given that the Polygon chain selected for this study employs a Proof-of-Stake (PoS) consensus algorithm, it can support a high transaction throughput [14]. Additionally, due to the absence of mining competition, gas price[21] fluctuations are lower [33]. In the Mumbai testnet, the gas price has remained stable at 2.5 Gwei[22].

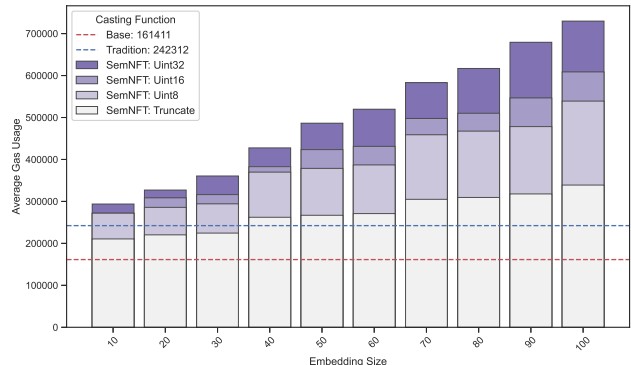

**Figure 10: Gas Usage with Different Casting Function and Embedding Size**

In each group of embedding sizes, we selected five embeddings for the minting experiments and introduced an additional group with an Embedding Size of 0 to serve as a Mint Baseline (without any extra data like link string or digit) for comparisons. We employed four different casting methods for various embedding sizes. Moreover, given the use of the same Polygon Mumbai testnet, both works benefited from consistent and stable gas prices. We compared the gas with the results from the Web3DP experiment [34], which represents the existing NFT minting process (with asset link string). The comparative results are illustrated in Figure 10.

We observe that as the Embedding Size increases, the required gas usage also gradually increases. However, it is worth noting that due to the EVM's optimized ability to compress array storage, the gas used related to Embedding does not increase proportionally with the increase in Embedding Size, particularly evident when comparing Embedding Sizes of 10 and 100.

When comparing the gas usage for existing NFT mints, it's important to consider that existing NFTs typically involve storing the IPFS link string, which is approximately 80 characters in length, on the blockchain. As a result, the gas usage for existing NFTs is slightly higher than that for mints that don't involve any storage. However, the gas for existing NFTs is comparable to that of mints with an embedding size of 40 and the Truncate casting method.

## 6.4 Storage Efficiency Comparison

As a result of varying Embedding Size in storage, discrepancies exist between the on-chain data size and the overall project's data size. Consequently, based on Equation 6 and 7, and without considering EVM's spatial optimizations and fixed model size, we have chosen

---

[21]At 21:55 On October 12, 2023, the standard gas price on the Polygon mainnet is 120 Gwei. The price of 1 Matic (MATIC) on Coinbase is $0.7.
[22]1 Gwei = $10^{-9}$ MATIC

**Table 3: Some data sizes and ratios among the existing NFT and SemNFT with embedding size 10 and 100.**

| Embedding Size | Existing | 10 | 100 |
|---|---|---|---|
| Single Data Size | 183KB | 30 Bytes | 300 Bytes |
| Model Size | - | 86MB | 86MB |
| On-chain Size | 0.8MB | 0.3MB | 3MB |
| Off-chain Size | 1.02GB | 86MB | 86MB |
| Total Size | 1.02GB | 86.3MB | 89MB |
| Compression Ratio | - | 11.82:1 | 11.46:1 |

the *Azuki* dataset and employed the UInt16 casting method, to analyze the spatial variations in on-chain data size and the overall project's data size when different Embedding Sizes are employed. The findings are presented in Table 3.

In our investigation, we conducted a comparative analysis between conventional NFT storage methods and scenarios involving embedding sizes of 10 and 100. SemNFT has exhibited a significant reduction in off-chain storage requirements. Notably, for embedding sizes of 10 and 100, we achieved compression ratios of 11.82:1 and 11.46:1, respectively. The demonstrated autoencoder model showcases superior compression performance and reduces the overall storage burden. Simultaneously, it offers customized options to achieve better image restoration effects while slightly exceeding the storage size of existing methods. In summary, it displays the efficiency of SemNFT in data compression and storage.

## 7 CONCLUSION

This paper presented SemNFT, a decentralized framework tackling pressing NFT storage and verification challenges. By leveraging decentralized middleware services equipped with autoencoders, the framework ensures not only the secure and perpetual storage of digital assets in the form of NFTs but also maintains their intrinsic qualities and representations, thereby safeguarding their essence throughout the pipeline. Our framework is also highly compatible with existing NFT standards, adapting well to the current NFT community and ecosystem. To verify its feasibility, we implemented the proposed framework and conducted an evaluation Evaluations evidenced feasibility and efficiency, attaining over 10:1 data compression ratio across diverse NFT datasets with high visual fidelity. SemNFT streamlines verification by reconstructing images on-demand, circumventing external dependencies. SemNFT emerges as a pioneering framework, intertwining the sophisticated capabilities of autoencoders with the immutable and decentralized characteristics of blockchain to forge a path towards digital asset immortality. Future research might delve deeper into optimizing the autoencoder algorithms, exploring additional use-cases for the framework, and evaluating the real-world impact and applications of ensuring digital asset immortality within various industries.

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
