# OpenReview forum: "SemNFT: A Semantically Enhanced Decentralized Middleware for Digital Asset Immortality"
_acmmm.org/ACMMM/2024/Conference — MM2024 Poster_

### Official Review · Reviewer_q8eU · 2024-05-10

**Rating:** 4
**Confidence:** 4

**Summary:**

Current NFT solutions utilizes on-chain and off-chain collaboration to authenticate ownership of unique digital content, which is challenged by the data availability that off-chain storage will remove original data after transactions. This paper utilizes autoencoder and decoder to compress NFT source data, and integrates them into blockchain oracle to achieve verification.

**Strengths:**

This paper provides a detailed implementation of the autoencoder enabled NFT interaction framework, including the adaption of smart contract and verification.

I consider the integration of blockchain and autoencoder (maybe I can use the term semantic communication to replace it) to be an interesting and important research question.

The paper is fairly well-written and is well-organized. The paper includes some critical figures that help to explain the problem and the proposed solution.

**Limitations:**

The blockchain oracle should be integrated into the network architecture. Moreover, how to achieve consensus on blockchain oracles is not well explained since the reconstructed images may be different in oracle nodes, which is important to this paper.

Since the reconstruction requires the encoder and decoder part should have similar knowledge, how to train and deploy the encoder and decoder should be well explained which is important to this paper.

This paper should also solve the issue that the blockchain oracle may not store the decoder model, which challenges the data availability again.

This paper is similar to [1] which also utilizes blockchain oracle and semantic verification to reduce storage costs. The authors should highlight the differences.

The experimental parts should at least compare the other two blockchain and semantic-enhanced storage methods. The current manuscript only compares the solution without semantic essence.

[1] A Unified Blockchain-Semantic Framework for Wireless Edge Intelligence Enabled Web 3.0.

**Suitability:**

3

---

### Official Review · Reviewer_j3qf · 2024-05-21

**Rating:** 4
**Confidence:** 2

**Summary:**

This paper proposes a semantically enhanced decentralized middleware framework called SemNFT, to address the existing NFT dilemmas. SemNFT brings an autoencoder into the framework to reduce the gas costs and the storage burden on-chain, by distilling the semantic essence of NFT images. The authors conduct evaluations of the proposed framework to prove its feasibility and efficiency.

**Strengths:**

-- The paper proposed a compatible method to solve the NFT secure and reliable storage problem.

-- The proposed method achieves good results according to the evaluation part.

-- The paper is well-organized with a clear and effective presentation.

**Limitations:**

-- The authors introduce the use of an autoencoder to reduce storage costs by providing a semantic embedding of NFT images through encoding and decoding. However, it seems that this work primarily applies the autoencoder methodology to NFTs, leveraging the inherent advantages of the method. To strengthen the paper, the authors should clearly articulate the unique contributions and innovations of their proposed method.


-- Although the proposed method effectively addresses the on-chain storage problem, the introduction of encoding and decoding operations incurs an additional computational burden. The authors need to analyze this overhead further and consider the practical user case.

**Suitability:**

2

---

### Official Review · Reviewer_pTWs · 2024-05-24

**Rating:** 2
**Confidence:** 2

**Summary:**

The SemNFT article presents a Semantically Enhanced Decentralized Middleware for Digital Asset Immortality. Its purpose is to address the storage and verification challenges faced by Non-Fungible Tokens (NFTs).

**Strengths:**

1. The article introduces the SemNFT framework, which is an innovative decentralized middleware solution addressing challenges in NFT storage and verification.

**Limitations:**

1. The formatting in lines 50-58 of the article needs correction.
2. The paper focuses primarily on image-based NFTs, with a claim that the framework can be extended to other types of NFTs. However, there is a lack of in-depth discussion or empirical evidence supporting the generalizability of the proposed framework to different types of NFTs.
3. The overall organization and logical flow of the paper appear to be average.
4. While the article mentions the importance of decentralization for security and reducing reliance on private servers or IPFS, it lacks in-depth discussion on specific security measures implemented within the SemNFT framework. Details on encryption methods, data protection mechanisms, and resistance to potential attacks are not extensively covered.

**Suitability:**

2

---

### Official Review · Reviewer_bPzE · 2024-05-24

**Rating:** 4
**Confidence:** 2

**Summary:**

In this paper, authors propose a decentralized framework integrated with blockchain oracle middleware services, called SemNFT, addressing NFT storage and verification challenges. SemNFT solves the high-cost storage problems by compressing NFT image data into compact embeddings and storing these arrays on-chain through smart contracts. SemNFT solves verification problems by facilitating reliable decentralized image reconstruction. The experiments are conducted from different perspectives, including reconstruction results and storage costs under different parameter settings.

**Strengths:**

1.	It is novel to apply image compression and reconstruction methods in the NFT domain for distributed ownership verification to achieve digital asset immortality.
2.	The proposed framework is well motivated and supported by both theoretical analysis and empirical evaluation. It has good adaptability with the Ethereum NFT protocol and uses the packing mechanism to optimize data compaction in Ethereum.
3.	This paper is overall well-written and organized.

**Limitations:**

1.	Advantages over the traditional approach (using IPFS) must be discussed in more detail.
The NFT ownership verification process is shown in Figure 7. The authors mention that IPFS may not be able to find the NFT image due to the metadata records disappearing during the process of querying/generating NFT images from the contract. SemNFT does not have this risk and is superior to traditional methods. However, the author does not explain the failure of the comparison step in Figure 7 due to image reconstruction distortion. Is there a situation where the probability of failure due to reconstruction graph distortion is greater than the probability of failure due to metadata loss in the IPFS scheme?
It is hoped that the author can add more details on the process of comparison and whether it will be affected by reconstruction distortion. It is necessary to compare the ability of the traditional method and SemNFT in the whole verification process.

2.	The trustworthiness of the middleware blockchain oracle and IPFS in the traditional approach must be compared.
In line 571, the blockchain oracle as middleware is regarded as trusted and is used as the platform for the model to run. But whether the distributed blockchain oracle has the same problem of data loss as IPFS (the problem of model data corruption and loss).

3.	Although the compression ratio is 11:1, based on the Figure 10 result, to ensure the quality of the reconstructed image, the uint8 or higher precision is required, and the gas overhead at this precision (purple part) is higher than that of the traditional method (blue dashed line).

**Suitability:**

3

---

### Meta-Review · Area_Chair_Leqp · 2024-07-01

**Recommendation:** Accept (Poster)
**Confidence:** 5

**Metareview:**

4 reviews completed with 1 weak reject, 2 weak accepts, 1 borderline accept; Area Chair concurs with majority of reviewers for accept given the importance of the topic of “authenticated ownership of unique digital content”.